# Electroassisted Incorporation of Ferrocene within Sol–Gel Silica Films to Enhance Electron Transfer

**DOI:** 10.3390/molecules28196845

**Published:** 2023-09-28

**Authors:** Rayane-Ichrak Loughlani, Alonso Gamero-Quijano, Francisco Montilla

**Affiliations:** Departamento de Química Física and Instituto Universitario de Materiales de Alicante (IUMA), Universidad de Alicante, Carretera San Vicente s/n, 03690 Alicante, Spain; lri11@alu.ua.es (R.-I.L.); daniel.gamero@ua.es (A.G.-Q.)

**Keywords:** silica functionalization, UV-vis spectroelectrochemistry, mediated electron transfer, cytochrome c

## Abstract

The sol–gel method is a straightforward technique that allows electrode modification with silica thin films. Furthermore, the silica pores could be functionalized to enhance the electrical conductivity and reactivity of the silica films. In this context, silica thin films were functionalized with ferrocene species. This functionalization was performed by electroassisted accumulation, generating a micro-structured composite electrode (Fc@SiO_2_ electrode). These modified electrodes were characterized by electrochemical and spectroelectrochemical methods, pointing out that ferrocene species were confined with high stability within the microporous silica thin film, demonstrating the good adsorption capacity of the silica. While the spectroelectrochemical characterization indicates that only a fraction of the confined species within the silica films were electroactive, the electrochemical results demonstrate that the Fc@SiO_2_ film enhances the electrochemical response of cytochrome c in a solution, which gives rise to further applications of these films for redox-controlled release and electrochemical detection of other redox-active proteins.

## 1. Introduction

Silica sol–gel films present attractive properties as permeable membranes for molecules or ions. The permeability of the film depends on its structure, thickness, and composition, as well as the size and charge of the molecules or ions involved. These films are straightforward to prepare, biocompatible, present good adsorption capacity [1,2], and have a large surface area [3,4,5]. Such properties could be advantageous in accumulating electroactive analytes within the film before electrochemical detection, enhancing the detection sensitivity and response time [6].

By modifying the sol–gel formulation or incorporating additional components, such as additives or surface functionalization, the permeability of the silica sol–gel film can be tailored to specific electrochemical applications. For example, the addition of organic or inorganic species can create nanoporous structures within the film, enabling size-selective filtration or separation of molecules. On the other hand, the film’s pores can be covalently grafted at such a level that they act as molecular sieves, improving electrochemical detection and crucially avoiding electrode fouling [7].

While silica films exhibit valuable features, their modification with active species boosts their performance and functionality. On the one hand, silica films can be modified with active agents such as metal oxides, metal cyanide derivatives, zeolites, polyoxometalates, and conducting polymers, among others [8,9,10,11]. These catalytic components can promote specific reactions, increase reaction rates, or improve selectivity. On the other hand, silica films can be modified with redox species that act as electron transfer mediators to different compounds such as photoactive species [12,13,14], redox probes [15], organic macromolecules (including biomolecules) [16,17], or microorganisms [18]. One of the most relevant molecules promoting effective electron transfer is the ferrocene/ferrocenium redox couple. Ferrocene and its derivatives have been extensively used in electrochemical applications [19], particularly in second-generation biosensor devices due to their high stability in redox reactions and ease of synthesis [20]. Thus, they have been employed as electronic transducers in glucose sensors [21,22,23].

In most cases, ferrocene-functionalized silica films are covalently attached to the silica surface to avoid leaching to the solution [24,25,26,27]. By combining click chemistry and silica functionalization, ferrocene species can be efficiently and selectively attached to the silica film surface. Although this approach allows precise control over the film’s density and distribution of ferrocene groups, it demands several hours of synthesis steps and advanced skills in organic chemistry.

Herein, we investigated the electroassisted modification of microporous silica sol–gel films with ferrocene species. This single-step method aims to create a composite micro-structured electrode of high electrochemical activity and long-term stability, addressing the issue of leakage that often arises in many electrochemical applications. The electroassisted accumulation process was precisely controlled within silica films to achieve the desired contact between biomolecules in a solution and the electrode surface, enabling molecular wiring. As a proof of concept, the ferrocene–silica composites were optimized for achieving mediated electron transfer to the model protein, cytochrome c (Cyt c), since it plays a crucial role in the respiratory chain and recently has been suggested as a biomarker in chronic diseases (e.g., breast cancer) [28]. Cyt c is a water-soluble protein with a heme active site that can undergo reduction/oxidation to form ferrous (Fe^+2^)/ferric (Fe^+3^) oxidation states [29,30] and, therefore, is an excellent model molecule to demonstrate mediated electron transfer reaction with ferrocene–silica-modified electrodes.

## 2. Results and Discussion

### 2.1. Electrochemical Performance of Bare ITO Electrodes in Ferrocenium Solutions

Figure 1 shows the stabilized cyclic voltammogram of a bare ITO electrode immersed in a ferrocenium-containing solution in a Trizma buffer.

During the forward scan from 0.1 V toward more positive potentials, an anodic peak appeared centered at 1.06 V and was attributed to the oxidation of neutral ferrocene to ferrocenium cations. During the backward scan, the counter process was observed as a cathodic peak centered at 0.72 V. The peak-to-peak separation was 0.34 V, and the half-wave potential was 0.89 V. The large peak-to-peak separation is attributed to a slow heterogeneous electron transfer between the ITO electrode and the redox pair in the solution. The quasireversible signal in Figure 1 shows different anodic and cathodic peak intensities of 43.5 µA and 60 µA, respectively. This corresponds to an anodic/cathodic peak ratio of 0.72 due to the absence of equal concentrations of the redox couple species (ferrocene/ferrocenium) in the bulk solution. Since neutral ferrocene has low solubility in aqueous solutions (7 µg/mL [31]), it was not added to the bulk aqueous solution.

Experiments at different scan rates were carried out to estimate the diffusion coefficient of Fc^+^ in aqueous solutions (see Appendix A). The inset in Figure 1 shows the plot of the cathodic peak current vs. the square root of the scan rate. A linear region was observed up to 0.2 V s^−1^, indicating that the electron transfer process was controlled by diffusion. The diffusion coefficient of ferrocenium (D) was estimated using the Randles–Sevcik equation as D = 1.2 × 10^−6^ cm^2^ s^−1^. This experimental value agrees with the values of diffusion coefficients reported in the literature [32,33]. At higher scan rates (υ > 0.5 V s^−1^), deviation from the linearity was noticeable, most likely due to IR drop.

### 2.2. Electrochemical Performance of Silica-Modified Electrodes

Silica thin films are not electroactive but present good ion exchange properties due to their surface chemistry and porosity [34]. They can act as permeable membranes, enhancing or hindering the electrochemical response of redox-active species in solution [35]. A direct example is given in Figure 2, wherein the electrostatic interactions between ferrocenium species and the surface chemistry of the silica film enhance the local concentration of ferrocenium species compared to the bulk solution. Figure 2 shows the repetitive cyclic voltammetry of an ITO/silica-modified electrode immersed in a 0.03 mM ferrocenium solution.

Cathodic and anodic peak intensities increased stepwise with the repetitive cycling, indicating a continued accumulation of redox species within the silica matrix. The ferrocene/ferrocenium redox process presented a peak-to-peak separation of 0.34 V with a silica-modified ITO electrode. Here, the reversibility of the reaction was similar to the bare ITO electrode (vide supra), meaning that despite the silica film being an electronic insulator, it does not affect the electron transfer kinetics. Note that in all the voltammograms, the anodic/cathodic intensity peak ratio was 0.95, near the theoretical ratio when both oxidized and reduced species concentrations were the same in the bulk solution. This suggested an improved ferrocene/ferrocenium concentration balance established at an ITO/silica-modified interface compared to bare ITO electrodes.

The electroassisted accumulation process within the silica matrices was studied using a sequence of different ferrocenium concentrations (0.06 to 0.36 mM). Repetitive voltammetry cycles showed a continued increase in the anodic and cathodic peak intensities, reaching a stabilized response after 60 scans in all the cases.

Figure 3A shows the final stabilized cyclic voltammograms of an ITO/SiO_2_ electrode after repetitive CV scans with different ferrocenium concentrations in a solution of 0.06 (red line), 0.09 (dashed line), and 0.12 mM (dotted line). Interestingly, the stabilized anodic and cathodic peak intensities increased 2.53-fold with 0.06 mM, which is above the expected 2-fold value by increasing the concentration from 0.03mM to 0.06mM. Similarly, the peak intensities with 0.09 mM ferrocenium increased four-fold, above the expected three-fold value.

Figure 3B shows the stabilized cyclic voltammograms of a SiO_2_-modified electrode immersed in 0.12 (solid line) and 15 mM (dashed line) ferrocenium solutions. Note that the increase in the concentration of ferrocenium from 0.12 mM to 0.15 mM led to an increase in the redox signal and the appearance of a second anodic peak centered at 1.33 V. However, a single cathodic peak centered at 0.59 V was observed during the reverse scan. The cathodic peak potential did not shift and continued the same trends as at lower concentrations, meaning that the reduction reaction (ferrocenium reduction) was unaffected by changes in ferrocenium concentrations in the bulk solution.

The stabilized cyclic voltammograms of SiO_2_-modified electrodes at higher ferrocenium concentrations (0.18 mM, 0.27 mM, and 0.36 mM ferrocenium) are shown in Figure 3C. The anodic peaks centered at 0.94 V and 1.33 V were more evident at concentrations above 0.15 mM, whereas a single cathodic peak centered at 0.62 V was recorded during the reverse scan. The anodic peaks overlapped for ferrocenium concentrations above 0.36 mM (see Figure 3C dotted line). It was evident that the anodic peak shape was concentration-dependent, presenting major changes at higher ferrocenium concentrations. Multiple anodic peaks have already been observed with ferrocene-functionalized mesoporous silica thin films and ferrocene-grafted mesoporous silica beads [24,25]. These changes were attributed to mass transport restrictions to maintain the electroneutrality within the negatively charged silica films, i.e., charge compensation limitations between the formed ferrocenium cations and the supporting electrolyte anions.

Furthermore, we carried out experiments at different scan rates, confirming the presence of two different kinetics of heterogeneous electron transfer attributed to mass transport limitations within the film (see Appendix A). At low scan rates (10*–*20 mV s^−1^), the CVs presented a single bell-shaped peak that could be interpreted as adsorbed ferrocene species on the ITO-SiO_2_ electrode interface. The diffusion layer thickness is wider at low scan rates, and mass-transport limitations within the silica film were not so evident voltammetrically. Furthermore, the intensity of the anodic peak at low scan rates presented higher intensities than the cathodic peak, suggesting the great affinity between the silica films and ferrocenium species. At 50 mV s^−1^, the peak started to progressively lead to a peak-to-wave change, as shown in Appendix A. At higher scan rates (100 mV s^−1^ to 1000 mV s^−1^), the mass transport is restricted by the silica film porosity, and, therefore, different electron transfer kinetics should be expected. Our silica films presented bimodal pore size distributions of 1.5 nm and 7 nm on average, estimated by N_2_ adsorption isotherms (further details are given in Appendix A and Appendix A).

Finally, Figure 3D shows the cathodic peak intensities recorded at different ferrocenium concentrations using SiO_2_-modified and bare ITO electrodes, respectively. Furthermore, we have included the theoretical peak intensity value expected for a diffusion-controlled redox reaction for comparison. The slope of the SiO_2_-modified electrode plot (red square) was 3-fold higher than bare ITO. Furthermore, the SiO_2_-modified electrode plot follows linearity even at high ferrocenium concentrations, stressing silica films*’* well-known anti-fouling effect [36]. In contrast, the bare ITO electrode presented a significant deviation from linearity at 0.24 mM (further details are given in Appendix A). From 0.03 mM to 0.24 mM, bare ITO electrodes (blue circles) presented a similar slope as the theoretical estimations (black triangles), i.e., the cathodic current was proportional to the bulk ferrocenium concentration. This study verified an accumulation effect of ferrocene/ferrocenium species occurring within the silica matrices. The accumulation is due to electrostatic interaction between a negatively charged silica and ferrocenium species. Here, the pH of the aqueous solution (pH 8.4) plays a major role, as it determines the surface chemistry and the surface charge of silica matrices. The potential of zero charge for silica/aqueous interfaces has been reported to be between pH 2 and 4, and, therefore, silica matrices under this pH must present a negatively charged surface [37]. Note that using mild alkaline conditions (pH 8.4) enhances the load of cationic species within the silica films [38]. pH is another parameter that can be modulated where a combination of electrostatic interactions, counterion release, and van der Waals forces enhance the electroassisted accumulation of species within the silica film [39].

### 2.3. Electrochemical Performance of Fc@Silica-Modified Electrodes

Silica films were functionalized by repetitive cyclic voltammetry cycling, accumulating ferrocene species and following the same experimental procedure described in Figure 2 but using a ferrocenium concentration of 0.30 mM. After obtaining a stabilized cyclic voltammogram, the electrode was removed from the electrochemical cell and rinsed with abundant ultrapure water. The modified electrode was immersed in a ferrocenium-free buffer solution. The electrochemical performance of the Fc@Silica-modified electrode was tested by repetitive cyclic voltammetry cycling between 0.1 and 1.8 V at 100 mV s^−1^, as shown in Figure 4.

The anodic and cathodic peak intensities decreased stepwise throughout the voltammetric cycling due to the leaching of the ferrocene species to the blank solution. After 40 cycles, a stable signal was reached (see dotted line in Figure 4), meaning that the electroassisted accumulation of ferrocene may be limited to ferrocene species remaining confined within the silica films. The apparent ferrocene concentration within the silica calculated from the reduction peak intensity of the stabilized cyclic voltammogram (dotted line) was 0.12 mM.

UV−vis experiments were carried out using a dried Fc@SiO_2_-modified electrode to confirm the chemical composition of the synthesized electrodes. Therefore, the ITO/Fc@SiO_2_-modified electrode was prepared as indicated in Section 2.2 and Section 2.3 and dried for one day at room temperature. The corresponding UV-Vis spectrum is presented in Appendix A. The Fc@SiO_2_-modified electrode exhibits a single absorption band at 620 nm, which can be attributed to the ligand-to-metal charge transfer band of ferrocenium (Fc^+^) [40] that is tightly confined within the silica matrix. Control experiments confirmed the ferrocenium band assignment (see Appendix A). It is worth noting that the silica thin film color transition from transparent to blue visually confirmed the presence of ferrocenium species (see Appendix A).

In situ UV−vis experiments were carried out at constant potential to explore the electrochemical behavior of encapsulated ferrocene/ferrocenium species within the silica films. Figure 5 shows the in situ UV-vis spectra of a Fc@SiO_2_ electrode prepared by repetitive cyclic voltammetry cycling, as shown in Figure 4. At 1.42 V, the UV-vis spectrum presented an absorption band centered at 620 nm, which was attributed to the ligand-to-metal charge transfer band of ferrocenium present within silica [40].

After applying a potential of +0.22 V vs. RHE (where ferrocenium species should be fully reduced), the stable UV-vis spectrum obtained (dashed line in Figure 5) shows that the ferrocenium band intensity partially decreased. It confirms that only a fraction of ferrocene–ferrocenium species are electroactive within the silica film, and they are likely confined in a small volume at the interface between the silica film and the ITO electrode. We investigated the surface chemical composition of both SiO_2_ and Fc@SiO_2_ films using X-ray photoelectron spectroscopy (XPS) (see Appendix A). The interpretation of the XPS spectra suggests the absence of iron within the Fc@SiO_2_ film. According to the literature, the Fe2p3/2 component is normally found in the binding energy (BE) range (709–711) eV [41]. Given the surface-sensitive aspect of XPS, this absence could imply that the accumulated ferrocene species are confined within the inner layers of the dried silica films.

A calibration curve was constructed to establish a quantitative relationship between the measured absorbance and the concentration of the confined ferrocenium species within the silica film (see Appendix A). The spectrum analysis where ferrocene species are fully oxidized indicated the presence of 1.37 mM of the ferrocenium species confined within the silica film. The spectrum analysis obtained after reduction indicated the presence of 1.18 mM of the confined ferrocenium species within the silica film. Therefore, from the difference between both values, we can determine that only around 0.19 mM of the confined molecules are electroactive (can change their redox states).

A large concentration of ferrocenium molecules stay in their original state, probably because they are tightly bonded to the silica surface and cannot diffuse to the ITO surface. The variation of concentration measured by UV spectroscopy is in good agreement with the value of concentration obtained by cyclic voltammetry (0.12 mM) since voltammetric measurement only allows for the determination of electroactive ferrocenium. The electrochemical decomposition of ferrocenium species was ruled out since no band shifting was observed [42].

### 2.4. Electrochemical Performance of Fc@Silica-Modified Electrodes for Electron Transfer to Cyt c

To assess and quantify the charge transfer signals generated by Fc/Fc^+^ and Cyt c- Fe (II)/Cyt c- Fe (III), the Fc@SiO_2_-modified electrode was prepared as indicated in Section 2.3. Subsequently, the electrode was immersed in a PBS buffer aqueous solution. Figure 6 shows the stabilized cyclic voltammogram of Fc@SiO_2_-modified electrodes in the presence and absence of Cyt c species.

The blank voltammogram (dotted line) shows a quasireversible signal at 1.13 V and 0.41 V, attributed to the redox response of the confined ferrocene–ferrocenium species within the silica film. The electrochemical response of the Fc@SiO_2_ electrode in a Cyt c-containing solution shows well-defined redox processes attributed to the redox-active protein. Two anodic peaks appear centered at 0.89 V and 1.13 V during the forward scan. The wave at 0.89 V corresponds to Cyt c-Fe (II) oxidation, whereas the second anodic peak centered at 1.13 V corresponds to the oxidation of Fc^+^ species confined within the silica film. During the backward scan, a single cathodic peak appeared centered at 0.41 V, suggesting that both ferrocenium and Cyt c are electrochemically reduced at this potential. The ferrocene–ferrocenium species trapped within the silica matrices act as molecular transducers improving electron transfer to Cyt c. Note that the electrochemical response of the Cyt c SiO_2_-modified electrode was poorly defined, as shown in Appendix A, and, therefore, the presence of Fc species within the film enhanced the redox response of Cyt c. The charge transfer under the cathodic peak current has increased by around 28.4 µC compared to the one recorded in the blank solution, demonstrating the electron transfer between encapsulated ferrocene and Cyt c species. The film*’*s electroactive ferrocene/ferrocenium species act as an electron shuttle between the ITO electrode and Cyt c species in bulk.

The formal reduction potential of ferrocenium/ferrocene vs. RHE was estimated at 0.82 V, whereas Cyt c presented a formal reduction potential of 0.64 V; therefore, some redox cycling reactions at the electrode surface could be expected. Here, the reduction potential induces Cyt c- Fe (II) formation, which can produce the electrochemical reduction in ferrocenium species Fc^+^ within the film, and the chemically regenerated ferrocene is re-oxidized at the electrode silica interface. Indeed, the formal potential shifted toward less positive potentials (see Figure 6 blue line), confirmed that a redox cycling reaction was occurring at the ferrocene–silica interface.[43,44] This redox cycling amplifies the redox current attributed to Cyt c species compared to a SiO_2_ electrode (see Appendix A). This sequence of electron transfer reactions is driven by the external biasing provided by the potentiostat, i.e., this potential-controlled ET reaction, and can be described as an electrochemical–chemical reaction (EC reaction).

## 3. Materials and Methods

The reagents used were tetraethyl orthosilicate (TEOS, Sigma-Aldrich, Burlington, MA, USA, reagent grade), ferrocenium hexafluorophosphate (97%, Sigma-Aldrich), cytochrome c from a bovine heart (95%, Sigma-Aldrich), ethanol (Sigma Aldrich), hydrochloric acid (Merck (Rahway, NJ, USA), 37%), Trizma base (99.9%, Sigma-Aldrich), nitric acid (65%, Panreac (Catalonia, Spain)), potassium nitrate (Merck, 99%), potassium dihydrogen phosphate, and dipotassium hydrogen phosphate (99%), which were from VWR Chemicals. All solutions were prepared using ultrapure water (18.2 MΩ cm) obtained from an ELGA lab water Purelab system.

Ferrocenium solutions were prepared in Trizma buffer (pH = 8.4) aqueous solutions. All the ferrocenium solutions were stored only for one day to avoid any instability issues reported in the literature [42]. Cytochrome c solutions (1 mg mL^−1^) were prepared by dissolving 1 mg of the protein in 1 mL of phosphate buffer solution (PBS = pH 7.3) made of K_2_HPO_4_ (0.15 M) + KH_2_PO_4_ (0.1 M).

The electrochemical experiments were performed using 3-electrode cells. The working electrode was indium tin oxide (ITO)-coated glass substrates (SOLEMS, ITOSOL30, sheet resistance 25–35 Ω). Before their use, the ITO glass was degreased by sonication in an acetone bath and electro-oxidized galvanostatically at 0.1 mA cm^−2^ for 1 min in a Trizma solution.

A reversible hydrogen electrode (RHE) was used as the reference electrode, and a platinum wire was used as a counter electrode. The oxygen was purged from the electrochemical cells by bubbling a N_2_ flow for 10 min before starting the experiments, and the N_2_ atmosphere was maintained during all the experiments. The electrochemical experiments were carried out using an ES161 eDAQ potentiostat equipped with an ED401 for data acquisition and controlled with Echart software. UV−vis spectra were acquired with a JASCO V-730 spectrophotometer using quartz cuvettes with 1 cm of the optical path. In situ UV−vis spectroscopy was performed using a custom-made UV-vis spectroelectrochemical quartz cuvette under a N_2_ atmosphere (see Appendix A).

For the preparation of electrodes, two stock solutions were prepared as follows:

Solution 1: The precursor for silica sol was prepared by mixing 3 mL of tetraethyl orthosilicate (TEOS), 4.1 mL of ethanol, and 2.9 mL of a 0.01 M hydrochloric acid (HCl) solution in a closed glass vial. The resulting mixture was magnetically stirred for one hour at room temperature.

Solution 2: A Trizma base aqueous solution with a pH of 8.44 was prepared by mixing a 0.1 M Trizma base aqueous solution with 0.1 M potassium nitrate (KNO_3_). To achieve the desired pH, concentrated nitric acid (HNO_3_) was added incrementally while monitoring the pH level.

For the electrode modification using a silica hydrogel film, solutions 1 and 2 were mixed within an Eppendorf tube in a 1:1 volume ratio. Subsequently, 40 μL of this resulting mixture was uniformly dispersed over 1.8 cm^2^ of a clean indium tin oxide (ITO) electrode. After a time period of 2–3 min, a homogeneous silica gel film was formed over the ITO surface. The sol–gel synthesis of the silica and the electrode modification with silica hydrogel film is presented in detail (see Appendix A). The estimated thickness of this silica hydrogel film under these experimental conditions was approximately 200 microns.

The porosity of the silica samples was determined through physical adsorption of N_2_ at −196 °C using an automatic adsorption system (Autosorb-6, Quantachrome). The samples were prepared as silica monoliths following the sol–gel method and were dried for 24h at 100 °C under vacuum. Pore size distributions were obtained by applying the DFT method to the adsorption isotherm using the software supplied by Autosorb- 6 (Quantachrome).

An X-ray photoelectron spectroscopy (XPS) analysis was carried out using a Thermo-Scientific K-Alpha spectrometer, which operates in a fully automated mode. The preparation of the ITO/SiO_2_-modified electrode followed the procedure outlined in Appendix A, while the ITO/Fc@SiO_2_-modified electrode was prepared in accordance with the methods outlined in Section 2.2 and Section 2.3. Subsequently, both electrodes were left to dry at room temperature for a period of one day.

## 4. Conclusions

The sol–gel-synthesized silica films demonstrated a substantial adsorption capacity for ferrocene species, possibly attributed to the presence of multiple pore sizes, including micropores and mesopores. Under the influence of an external bias, these silica films accumulated a notable 1.37 mM of ferrocenium species, exhibiting remarkable stability. The XPS analysis indicated that these accumulated ferrocene species are predominantly confined within the inner layers of the silica films. In situ UV-Vis spectroscopy further revealed that only a fraction, specifically 0.19 mM, of the confined molecules exhibited electroactivity. This observation suggests that approximately 1.18 mM of ferrocenium molecules remained in their original state, possibly due to their strong bonding to the silica surface, preventing their diffusion to the ITO surface. The results of these studies offer insight into the complex interplay between ferrocenium species*’* adsorption, confinement, and electroactivity inside the silica layer. We have demonstrated that Fc@silica films enhance the electrochemical response of cytochrome c (Cyt c) in a solution, overcoming the electrochemical limitations observed with SiO_2_-ITO electrodes. This straightforward method of encapsulating Fc species within silica films provides a promising approach for further electrochemical studies of biological species and has the potential to be extended to other proteins in aqueous solutions. In conclusion, this study highlights the potential of silica synthesized through the sol–gel methodology as a biocompatible material for electrochemical studies of biological species.

## Figures and Tables

**Figure 1 molecules-28-06845-f001:**
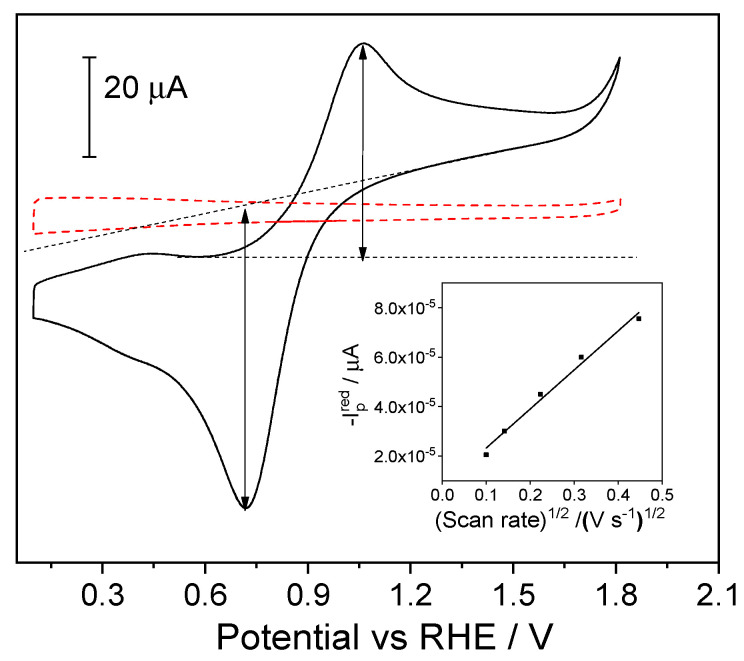
Stabilized cyclic voltammogram of an ITO electrode immersed in a 0.3 mM ferrocenium hexafluorophosphate (FcPF_6_) + Trizma buffer (solid line). The blank voltammogram of an ITO electrode in a ferrocenium-free solution is given in dashed lines. Scan rate: 100 mV s^−1^. Inset: Randles−Sevcik plot of the cathodic peak current vs. the square root of the scan rate.

**Figure 2 molecules-28-06845-f002:**
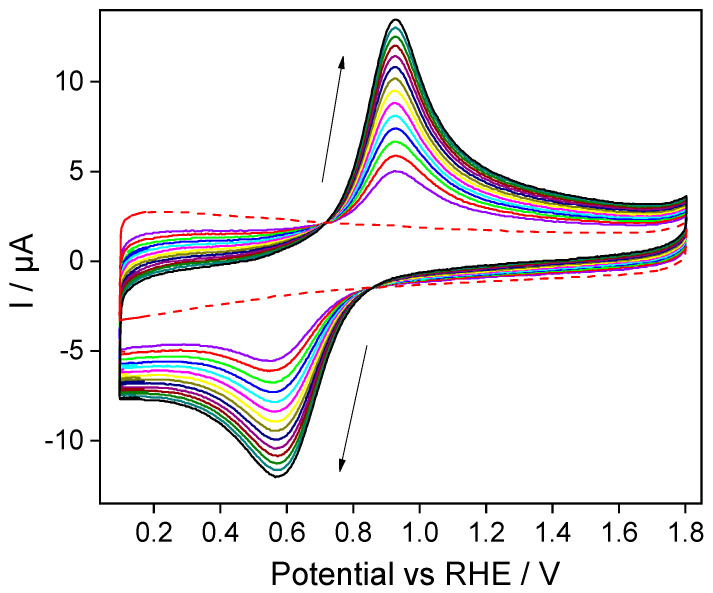
Repetitive cyclic voltammetry cycling of an ITO/SiO_2_-modified electrode immersed in a 0.03 mM ferrocenium + Trizma buffer. Arrows indicate the evolution of the current during successive cycles. The blank voltammogram of an ITO/SiO_2_ electrode in a ferrocenium-free solution is given in a dashed line. Scan rate: 100 mV s^−1^.

**Figure 3 molecules-28-06845-f003:**
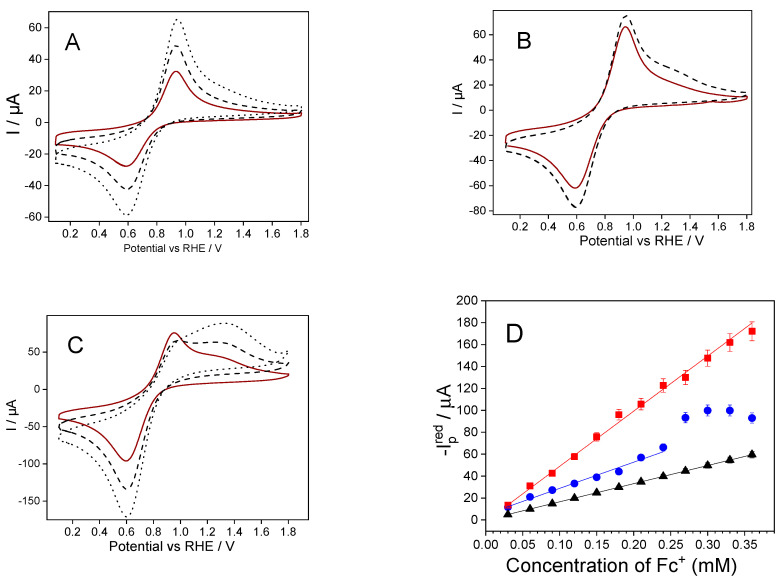
Stabilized cyclic voltammograms of a SiO_2_-modified electrode immersed in different concentrations of ferrocenium. Scan rate: 100 mV s^−1^. (**A**) A 0.06 mM solid line; 0.09 mM dashed line; 0.12 mM dotted line. (**B**) A 0.12 mM solid line; 0.15 mM dashed line. (**C**) A 0.18 mM solid line; 0.27 mM dashed line; and 0.36 mM dotted line. (**D)** Cathodic peak current vs. Fc^+^ concentration plot of the ITO electrode (blue circles), ITO/SiO_2_ electrode (red squares), and the theoretical values calculated using the Randles–Sevcik equation (black triangles).

**Figure 4 molecules-28-06845-f004:**
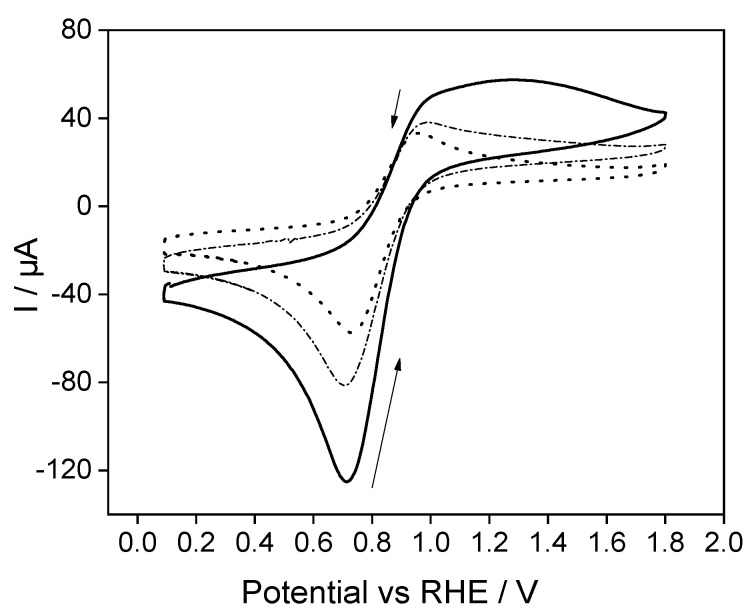
Repetitive cyclic voltammogram of a Fc@SiO_2_-modified electrode obtained in a Trizma buffer solution. Arrows indicate the evolution of the current during successive cycles. Solid line, first scan; dashed line, 5th scan; dotted line, 40th scan. Scan rate: 100 mV s^−1^.

**Figure 5 molecules-28-06845-f005:**
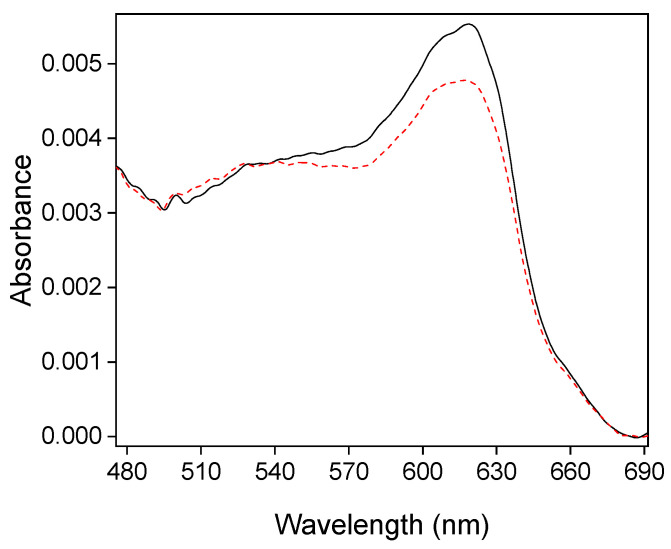
In situ UV−vis spectra collected for a Fc@SiO_2_-modified electrode. The solid line shows the spectrum obtained at +1.42 V vs. RHE (black solid line), whereas the dotted line shows the spectrum obtained at +0.22 V vs. RHE (red dashed line).

**Figure 6 molecules-28-06845-f006:**
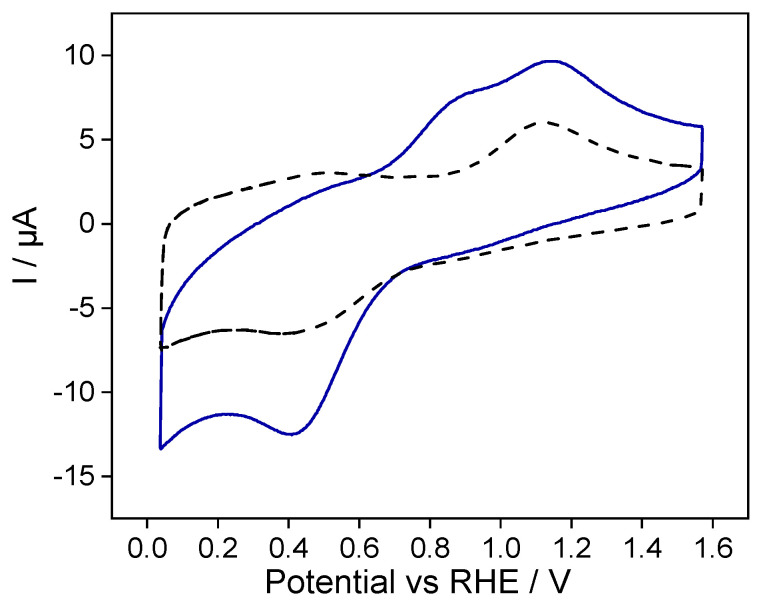
Stabilized cyclic voltammograms of Fc@SiO_2_-modified electrode (1 mg mL^−1^ Cyt c + PBS buffer aqueous solution) (solid line). The blank voltammogram Fc@SiO_2_-modified electrode immersed in the Cyt c-free solution is given in dashed lines. Scan rate: 100 mV s^−1^.

## Data Availability

The data presented in this study are available on request from the corresponding author.

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
