# Peer review of "Electroassisted Incorporation of Ferrocene within Sol–Gel Silica Films to Enhance Electron Transfer"

_molecules, 2023, doi:10.3390/molecules28196845_

Round 1

Reviewer 1 Report

The present submission “Electroassisted incorporation of Ferrocene Sol-Gel within Silica Films to enhance Electron Transfer” described the synthesis of a micro-structured composite electrode and corresponding response to Cytochrome C. Obviously, this study is significant, because this electrode can be used as sensor for detection of protein. And meanwhile, the electrochemical study of this work is comprehensive and sufficient. However, the most important drawback of this work is the lack of study on Structure-Activity Relationship. This part must be improved before acceptance. Therefore, I would like to recommend this work for publication in Molecules. However, before acceptance, I will have to give some suggestions that seem very important to improve.

(1)    The authors need to show the synthesis of composite electrode clearly in Sect. 3. Materials and Methods, particularly regarding sol-gel process.

(2)    The authors need to characterize the sol-gel product clearly, only nitrogen physisorption was not enough to illustrate the product. For example, whether ferrocene was changed during sol-gel, XRD and XPS are recommended for improvement.

(3)    How to evaluate the morphology and internal structure of synthesized electrode, SEM and TEM are recommended.

(4)    With the synthesized electrode and characterization, the authors need to illustrate Structure-Activity Relationship, and find out the key structural factors that affected the sensitivity of synthesized electrode.

    If the above issues can be improved, I would like to recommend this work for publication in Molecules.

The English presentation of this work is fine.

Author Response

Complete response (including figures) in the attached pdf

We thank the reviewers for their time in examining our manuscript and for providing their valuable comments.

We consider their opinion and suggestions to be crucial and have addressed them in full in the revised manuscript.

 In the following section, we have clarified and expanded below on all points raised and discussed in detail the corresponding changes made to the revised manuscript.

The most important points in our responses are underlined, and the corresponding corrections made in the main text/SI are highlighted in yellow.

Reviewer 1

Comment #1

The authors need to show the synthesis of composite electrode clearly in Sect. 3. Materials and Methods, particularly regarding the sol-gel process.

Response #1

We thank reviewer 1 for reading our manuscript and suggesting constructive modifications. The updated manuscript presents a more detailed Section 3. In the materials and methods section, we have given more details on the procedure of silica film formation on ITO electrodes. New paragraphs have been included in the revised manuscript and in the supporting information (highlighted in yellow).

Furthermore, regarding the sol-gel process, we have included a scheme describing the full procedure and the chemical reaction involved in the sol-gel process. representation of the sol-gel process and details regarding electrode modification in Fig. S11 in the supporting information.

Comment #2

The authors need to characterize the sol-gel product clearly, only nitrogen physisorption was not enough to illustrate the product. For example, whether ferrocene was changed during sol-gel, XRD and XPS are recommended for improvement.

Response #2

Reviewer 1 has suggested additional characterisation of the silica thin films by XPS and XRD analysis. First, we would like to remark that silica modified electrodes are in hydrogel state during the synthesis and in its application in the electrochemical measurement. However, these technique require ultra-high vacuum conditions and therefore samples must be dried before characterization. This drying step may modify the physico-chemical characteristics of the films. However, the measurements proposed by the reviewer are of interest and we consider that they improve the quality of our article.

We carried out additional measurements by XPS of two different samples a silica-modified electrode in absence of ferrocene (blank control) and a ferrocene-modified silica film. The differences between samples were noticeable by the change in the colour of the silica films, as shown in Fig.S5 B where the ferrocene-modified electrode presents a bluish colouration.

The XPS spectra are shown in Fig. S7, and the description is given in the SI section in the updated manuscript (see lines 263-269). Among the different elements measured by XPS it is noteworthy  the absence of iron signals for the Fc@SiO2 film. Since XPS is a surface-sensitive technique, this absence imply that the accumulated ferrocene species are confined within the inner layers of the dried silica films. However, the ferrocene is accessible for the solution when the silica is in hydrogel state as demonstrated by in situ UV-vis and CV experiments.

Furthermore, we carried out X-Ray diffraction studies (XRD), however, no conclusive differences were detected between modified and unmodified silica samples. In fact, no diffraction peaks were observed in any sample. Since our sol-gel method produces amorphous silica thin films without defined crystallinity and therefore the sample are characterised by the absence of XRD peaks.

Comment #3

How to evaluate the morphology and internal structure of synthesized electrode, SEM and TEM are recommended.

Response #3

We carried out TEM analysis for silica unmodified and modified samples to assess their morphology and internal structure. However, the micrographs do not provide clear differences between samples. The major conclusion is that an amorphous microstructured silica material was synthesised through the sol-gel method and ferrocene incorporation does not modify this aspect.

Fig. 1 shows the micrographs corresponding to the SiO2 and Fc@SiO2 films. The micrograph corresponding to SiO2 film (Fig. 1A) shows a densely packed amorphous structure. The size and shape of the porous structures within the SiO2 matrix are below the 10 nm mark in agreement with our BET analysis.  On the other hand, Fig. 1B shows the micrograph corresponding to the Fc@SiO2 films. The incorporation of ferrocene species within the film did not produce any morphological change.

Fig. 1. TEM micrograph corresponding to the (A) SiO2 and (B) Fc@SiO2films.

As no clear information is obtained from these measurements, we did not included it in the paper, but if reviewer considers we are open to include this data.

Comment #4

With the synthesized electrode and characterization, the authors need to illustrate the Structure-Activity Relationship, and find out the key structural factors that affected the sensitivity of synthesized electrode.

Response #4

We agree with the reviewer's concern regarding the need to illustrate the structure-activity relationship, since this is a critical aspect to address. Thus, we have extended the discussion and the conclusions of the revised manuscript (new paragraphs are highlighted in yellow).

Reviewer 2 Report

In this work, the authors have developed silica-thin films by electrodeposition as an electroactive material to promote electron transfer with Cyt C which resembles a mediating electron transfer mechanism. The work has been well structured and the results with the proof-of-concept with the Cyt C demonstrate the applicability of the electrode material. However, it is important that the authors clarify and answer some details:

1. What was the reason for using only the reduction peak of Fc/Fc+ redox couple for the Randles-Sevcik analysis?

2. I can understand the change of electrolyte from Trizma to PBS, but the adsorption of Fc/Fc+ species in Figure 6 was performed in the same electrolyte (PBS) or Trizma buffer. Authors should include that in the manuscript.

3. In line 225, the authors explain that the Fc/Fc+ signal during the "desorption" reaches a stable current density, thus, there are only ferrocenium species entrapped. Was it corroborated using XPS or FTIR measurements?

Author Response

Complete response  in the attached pdf.

We thank the reviewers for their time in examining our manuscript and for providing their valuable comments.

We consider their opinion and suggestions to be crucial and have addressed them in full in the revised manuscript.

 In the following section, we have clarified and expanded below on all points raised and discussed in detail the corresponding changes made to the revised manuscript.

The most important points in our responses are underlined, and the corresponding corrections made in the main text/SI are highlighted in yellow.

Reviewer 2

Comment #1

What was the reason for using only the reduction peak of Fc/Fc+ redox couple for the Randles-Sevcik analysis?

Response #1

First, we would like to thank the reviewer for raising concerns about important points that have been addressed in the updated manuscript.

The main reason for using the reduction peak of the cyclic voltammogram is because the Randles-Sevcik is applied only to diffusion-controlled redox processes. Our studies have been carried out in the presence of ferrocenium cations (Fc+) dissolved in the bulk solution, but no ferrocene species is present in the solution. Therefore, the most reliable reaction to consider was the reduction of ferrocenium species which is related to the cathodic peak that is proportional to the concentration of this compound.

Since ferrocene is absent in solution, the oxidation current detected comes only from the reduced ferrocenium on the electrode surface. Note that the bulk solution primarily contains only ferrocenium cations (Fc+). Furthermore, as section 2.2 (lines 165-170) explained, we observed a notable mass transport limitation with two different anodic peaks during the oxidation process meaning that the anodic peak could need more advanced mathematical tools beside the Randles-Sevcik equation to analyse the diffusion-controlled reaction.

Comment #2

I can understand the change of electrolyte from Trizma to PBS, but the adsorption of Fc/Fc+ species in Figure 6 was performed in the same electrolyte (PBS) or Trizma buffer. Authors should include that in the manuscript.

Response #2

We thank the reviewer for raising this concern regarding the buffers used in our manuscript. We want to clarify that the Fc@SiO2 modified electrode was prepared in Trizma aqueous solution, as indicated in section 2.3. A new paragraph has been added in section 2.4 (highlighted in yellow).

Furthermore, the cyclic voltammogram in dashed line shown in Fig. 6 was the blank voltammogram of the Fc@SiO2 modified electrode immersed in PBS buffer aqueous solution. Before immersing the Fc@SiO2 in cytochrome c solutions, a blank voltammogram was recorded to assess and quantify the charge transfer signals generated by accumulated Fc/Fc+ species. This blank CV was compared to the CV obtained in a Cyt c solution (see Fig. 6 blue line).

Comment #3

In line 225, the authors explain that the Fc/Fc+ signal during the "desorption" reaches a stable current density, thus, there are only ferrocenium species entrapped. Was it corroborated using XPS or FTIR measurements?

Response #3

We thank the reviewer for their expert advice. The presence of ferrocenium species confined within the silica film was demonstrated by the in-situ UV-Vis as shown in Fig. 5. When a positive potential of 1.42 V was applied, the UV-vis spectrum presented changes in the ferrocenium absorption band centred at 620 nm. The assignment of this band to ferrocenium species was validated through control experiments (see fig. S6 in the supporting information) Furthermore, we have carried out solid-state UV-Vis studies of the Fc@SiO2 modified electrode to support our finding; these new studies have been included in the updated manuscript in Fig. S5.

First, we would like to remark that silica modified electrodes are in hydrogel state during the synthesis and in its application in the electrochemical measurement. However, FTIR or XPS techniques require vacuum conditions and therefore samples must be dried before characterization. This drying step may modify the physico-chemical characteristics of the films. However, the measurement proposed by the reviewer are of interest and we consider that they improve the quality of our article.

We carried out additional measurements by XPS of two different samples a silica-modified electrode in absence of ferrocene (blank control) and a ferrocene-modified silica film. The differences between samples were noticeable by the change in the colour of the silica films, as shown in Fig.S5 B where the ferrocene-modified electrode presents a bluish colouration. Among the elements measured by XPS it is noteworthy  the absence of iron signals for the Fc@SiO2 film. Since XPS is a surface-sensitive technique, this absence imply that the accumulated ferrocene species are confined within the inner layers of the dried silica films. However, the ferrocene is accessible for the solution when the silica is in hydrogel state as demonstrated by in situ UV-vis and CV experiments.

Reviewer 3 Report

This manuscript on the electron transport properties of electro-assisted Ferrocene-incorporated Silica Sol-Gel thin films, which was unique in the idea of using Silica pores to introduce Ferrocene, as shown in Fig.3 and Fig.6, The changes in electrochemical properties were also captured. However, this paper lacked several necessary components.

First, in the discussion of Fig. 3, the authors discussed the change factors of the anodic peak intensity with respect to the concentration of the immersed Ferrocene solution, but was all of the immersed Ferrocene introduced into the Silica Sol-Gel thin film? If so, what method was used to confirm this? In addition, if the amount of Ferrocene introduced into the Silica Sol-Gel thin film was not clear, did it make sense to discuss the magnification factor in relation to the peak intensity? The reviewer could not understand how it could be said that “which presented good adsorption capacity of ferrocene species” in the “4. Conclusion”. What was a good adsorption property compared to?

Second, qualitatively, the introduction of ferrocene into the silica Sol-Gel thin films has changed their electrochemical properties. The authors claimed that Ferrocene was incorporated into the pores of Silica, but was there any evidence for this, and shouldn't SEM or TEM observations confirm the state?

A significant additional data will be required to address these queries, but the reviewer is looking forward to your good revision.

Author Response

Complete response (including figures) in the attached pdf

We thank the reviewers for their time in examining our manuscript and for providing their valuable comments.

We consider their opinion and suggestions to be crucial and have addressed them in full in the revised manuscript.

 In the following section, we have clarified and expanded below on all points raised and discussed in detail the corresponding changes made to the revised manuscript.

The most important points in our responses are underlined, and the corresponding corrections made in the main text/SI are highlighted in yellow.

Reviewer 3

Comment #1

First, in the discussion of Fig. 3, the authors discussed the change factors of the anodic peak intensity with respect to the concentration of the immersed Ferrocene solution, but was all of the immersed Ferrocene introduced into the Silica Sol-Gel thin film? If so, what method was used to confirm this? In addition, if the amount of Ferrocene introduced into the Silica Sol-Gel thin film was not clear, did it make sense to discuss the magnification factor in relation to the peak intensity? 

Response #1

First, we would like to thank the reviewer for dedicating time to reading our manuscript. We agree with the concern about the magnification factor related to peak intensity. Our work reports the electrochemically assisted accumulation of ferrocenium/ferrocene species at an ITO electrode modified with a porous silica film. This accumulation was induced, controlled, and monitored by repetitive cyclic voltammetry and it was found to be dependent on the ferrocenium concentration in solution.

 The effect of ferrocenium concentration was discussed in section 2.1. After several CV cycles , a stable signal was achieved, the electrode was then, extracted from the ferrocenium aqueous solution and was rinsed with abundant ultrapure water. It is important to remark that not all the ferrocenium in the solution was absorbed within the film, in fact the concentration remains practically constant during the adsorption experiments since the volume of the electrochemical cell was around 80 mL and the volume of the film was only around 0.04 mL, so the absorbed amount is negligible compared with the total amount of ferrocenium in solution.

After absorption stage, the electrode was transferred to a blank solution, where some of the accumulated species were lost in the solution but a stabilised value of peak current was achieved upon repetitive CV scans. Once a stable signal was obtained, we were able to determine the concentration of the electroactive ferrocenium/ferrocene species within the silica thin film.

Comment #2

The reviewer could not understand how it could be said that “which presented good adsorption capacity of ferrocene species” in the “4. Conclusion”. What was a good adsorption property compared to?

Response #2

We totally agree with the reviewer’s comment that the use of the sentence “silica films presented good adsorption capacity of ferrocene species” written in the conclusion section was not appropriate and has been removed from the updated manuscript.

Comment #3

Second, qualitatively, the introduction of ferrocene into the silica Sol-Gel thin films has changed their electrochemical properties. The authors claimed that Ferrocene was incorporated into the pores of Silica, but was there any evidence for this, and shouldn't SEM or TEM observations confirm the state.

Response #3

We thank the reviewer for their good insights, we would like to address that the introduction of ferrocenium species into the silica Sol-Gel thin films has already been confirmed by in situ UV-vis measurements described in Fig. 5 in the manuscript.

First, we would like to remark that silica modified electrodes are in hydrogel state during the synthesis and in its application in the electrochemical measurement. However, TEM or SEM analyses require ultra-high vacuum conditions and therefore samples must be dried before characterization. This drying step may modify the physico-chemical characteristics of the films.

In any case, we carried out TEM analysis for silica unmodified and modified samples to assess their morphology and internal structure. However, the micrographs do not provide clear differences between samples. The major conclusion is that an amorphous microstructured silica material was synthesised through the sol-gel method and ferrocene incorporation does not modify this aspect.

Fig. 1 shows the micrographs corresponding to the SiO2 and Fc@SiO2 films. The micrograph corresponding to SiO2 film (Fig. 1A) shows a densely packed amorphous structure. The size and shape of the porous structures within the SiO2 matrix are below the 10 nm mark in agreement with our BET analysis.  On the other hand, Fig. 1B shows the micrograph corresponding to the Fc@SiO2 films. The incorporation of ferrocene species within the film did not produce any morphological change.

Figure 1. TEM micrograph corresponding to the (A) SiO2 and (B) Fc@SiO2films.

As no clear information is obtained from these measurements, we did not include it in the paper, but if reviewer considers we are open to include this data.

Following the reviewer’s advice, additional UV-vis measurements have been performed and now are presented in Fig. S5 in the supporting information. Ferrocenium species within the films changed the electrode's colour from a colourless silica film to a bluish one. These changes were detected by UV-vis spectroscopy, shown in Fig. S5A. Furthermore, those optical changes were visible in the photograph of dried ITO/SiO2 and ITO/Fc@SiO2 modified electrodes shown in Fig. S5 B the supporting information.

The assessment of the morphology and internal structure of the synthesized electrodes through TEM analysis was carried out; however, we have decided not to include these results since no clear information was provided.

Round 2

Reviewer 1 Report

After revisions, I think this submission can be accepted. 

English writing is good and acceptable.

Reviewer 3 Report

Thank you for your careful revision of the previous manuscript

The reviewer confirmed that your appropriate revisions and not including no clear information of TEM observation.

This manuscript will be recommended to this journal.